# Self-Organized Anodic Oxides on Titanium Alloys Prepared from Glycol- and Glycerol-Based Electrolytes

**DOI:** 10.3390/ma13214743

**Published:** 2020-10-23

**Authors:** Marta Michalska-Domańska, Magdalena Łazińska, Justyna Łukasiewicz, Johannes M. C. Mol, Tomasz Durejko

**Affiliations:** 1Department of Materials Science and Engineering, Delft University of Technology, Mekelweg 2, 2628 CD Delft, The Netherlands; J.M.C.Mol@tudelft.nl; 2Institute of Optoelectronics, Military University of Technology, 2 Kaliskiego Str., 00-908 Warsaw, Poland; 3Faculty of Advanced Technology and Chemistry, Institute of Materials Science and Engineering, Military University of Technology, 2 Kaliskiego Str., 00-908 Warsaw, Poland; magdalena.lazinska@wat.edu.pl (M.Ł.); justyna.lukasiewicz@wat.edu.pl (J.Ł.)

**Keywords:** anodization, biomedical alloys, titanium anodic oxide, electrolytes viscosity, current density, oxidation susceptibility, ions incorporation

## Abstract

The anodization of commercially pure Ti alloy (99.5 wt %) and two biomedical titanium alloys, Ti6Al7Nb and Ti6Al4V, was performed, and the resulting anodic oxides were studied. The biomedical alloys were made by Laser Engineered Net Shaping. The glycol-based and glycerol-based electrolytes with 0.3 M ammonium fluoride and 2 wt % of deionized water content were tested. It was found that electrolyte type as well as the chemical composition of the base substrate affected the final morphology and chemical composition of the anodic oxide formed. A higher current density, ionic mobility, and oxide growth rate were obtained in glycol-based electrolyte as compared to those obtained in glycerol-based electrolyte for all tested alloys. A self-organized nanotubular and nanoporous morphology of the anodic oxide in both types of electrolyte was obtained. In each electrolyte, the alloy susceptibility to oxidation increased in the following order: Ti6Al4V < Ti 99.5% < Ti6Al7Nb, which can be correlated to the oxidation susceptibility of the base titanium alloy. It was observed that the more impurities/alloying elements in the substrate, the lower the pore diameters of anodic oxide. There was a higher observed incorporation of electrolyte species into the anodic oxide matrix in the glycerol-based electrolyte compared with that in glycol-based electrolyte.

## 1. Introduction

Anodization is a widely known method to low-cost and fast electrochemical surface treatment, which leads to modified surface properties or even allows obtaining new nanomaterials [1,2]. Titanium and its alloys are characterized by their high strength, low weight-to-durability ratio, and excellent corrosion resistance. In aerospace and biomedical applications, one of the most commonly used material is the Ti6Al4V alloy [3,4], but it contains vanadium, which can be considered harmful if released to organisms during long-term exposure [5]. On the other hand, there is a new perspective titanium alloy, Ti6Al7Nb, which contains non-toxic Nb and possesses even higher corrosion resistance and bio-tolerance than Ti6Al4V [5,6,7]. Generally, the high bio-tolerance of Ti alloys results from the spontaneous formation of a native passive layer on its surface of a few nanometers. Moreover, the recent research of implant–biological environment interfaces shows that the surface morphology, including oxide layer thickness, chemical composition, as well as crystallographic structure has a significant influence on the osteointegration process [8,9,10].

The effective way to increase the thickness of the oxide layer, and thereby improve implant surface properties, is to perform electrooxidation of the Ti alloy. The anodizing of Ti alloys is currently the subject of extensive research, particularly in terms of the assessment of optimum process conditions such as anodization voltage, time, or electrolyte for the generation of adherent oxide layers on the surface of titanium alloys designed for biomedical application [11]. The main advantages of anodic titanium oxides are ease of their formation directly on the surface of the material as well as the possibility to control the layer thickness by means of changing the process parameters [3,5,12]. Moreover, the ability to obtain a well-adherent nanoporous oxide layer could markedly facilitate the further development of nano-biomaterials based on Ti [13]. Nowadays, a lot of efforts is performed to develop an anodizing window with optimal parameters for biomedical Ti alloys [7,14]. For biomedical application, the fabrication of thicker and stable oxide layer on the top of implant are important because it is favorable for surface bioactivity to enhance bone-forming function and to facilitate osteointegration [11,15,16].

Traditionally, titanium was anodized in solutions based on HF in water, but the anodic oxide obtained as such was poorly ordered, thin, and rich in defects. For over a decade, viscous electrolytes containing glycol, water, and ammonium fluoride have been developed [17,18], providing the opportunity to obtain thicker titanium oxides. Generally, in water-based electrolyte contained HF, nanotubes with a length of about 500 nm can be obtained, while in more viscous electrolyte containing fluoride salts, smooth walled nanotubes with lengths of several micrometers can be produced [19]. Depending on the experimental condition, in the ethylene glycol-based electrolyte, it is possible to fabricate smooth-walled nanotubes of anodic titanium oxide with an aspect ratio of up to 32, while the nanotube made in the glycerine-based electrolyte can reach an aspect ratio of up to 150 [20]. The reason is that in glycerine-based electrolyte, the current efficiency for anodic oxide nanotubes formation is close to 100%, which is higher than in glycol-based electrolyte (<100%) and significantly higher than in water-based electrolyte (5–40%) [20]. Recently, we have reported a new electrolyte type for titanium anodization, which resulted in titanium anodic oxide morphology placed between anodic titanium oxide (ATO) fabricated in water-based and viscous electrolyte: ripped-walls nanotubes with lengths between 1 and 1.35 µm and diameters [21].

Anodization conditions are important factors allowing the control of anodic oxides morphology where, among others, the electrolyte viscosity has a significant influence on the anodization process and the resulting oxides [20,22,23,24]. The electrolyte viscosity is inversely proportional to the ionic mobility and the electrical conductivity of the electrolyte [22,25]. Moreover, the current density during the electrochemical processes is proportional to ionic mobility, and in liquid solutions, the reactions are often controlled by diffusion, while the diffusion coefficient strongly depends on ionic mobility [22,25]. The change of electrolyte viscosity influences the current density and diffusion coefficient, so the changes in the growth and properties of anodic aluminum oxide (AAO) are natural consequences of viscosity change. As was reported in one of our previous papers [22,23,24], in the case of aluminum anodization, (1) the current density and thickness of the formed oxide are linear functions of the inversed viscosity, (2) the interpore distance is a linear function of the logarithm of the dynamic viscosity, (3) the pore diameter is decreased with the increased electrolyte viscosity, and (4) higher electrolyte viscosity allows applying higher potentials as well as enlarging the temperature range possible to apply for anodization process in comparison to aqua-based electrolyte solutions. In the case of titanium anodization, the change of electrolyte viscosity has an exceptionally strong influence on the anodic titanium oxide morphology. Thanks to changing the electrolyte solvent from water to glycol or glycerine, it is possible to fabricate smooth-walled nanotubes of anodic titanium oxide with an aspect ratio increased from ≈3.5 obtained in water-based electrolyte [26] up to 32 and 150 in glycol-based and glycerine-based electrolyte, respectively [20]. 

A novel potentially increasing production route to improve the quality of the osteointegration process between bones and implants is the production of exclusive, specially designed individual implants by additive manufacturing technology. The Laser Engineered Net Shaping technique (LENS^TM^) is an additive manufacturing technology that involves metal powder being injected directly into the high-powered laser focus with the element of predefined geometry being constructed in a layer-by-layer fashion [27]. It is possible to fabricate alloys with strictly defined shape, composition, and structure [28,29] as well as a bespoke surface [30]. It is possible to produce metal part/implants directly from a computer-aided design (CAD) solid model, which opens the way to the fabrication of individual and dedicated implant designs. 

The main aim of the presented research is the investigation of the influence of the anodizing electrolyte composition on the morphology and chemical composition of the titanium anodic oxide layer formed on two LENS-made biomedical titanium alloys. 

## 2. Materials and Methods 

Ti6Al7Nb and Ti6Al4V alloys were used as the base substrate. A 99.5% commercially pure titanium sheet (Alpha Aesar) was used as a reference material. The Ti6Al4V and Ti6Al7Nb alloys cylindrical samples (φ = 20 mm, h = 50 mm) were fabricated by an LENS MR-7 system from alloyed, gas atomized, spherical titanium powders with a particle size in the range of 45–115 μm purchased from LPW Technology (Philadelphia, PA, USA). The CADmodel of the sample was sliced into 166 layers with a thickness of 0.3 mm. The deposition lines were 0.4 mm apart. The LENS process was conducted at 280 W of power laser, 16 mm/s of traverse speed, and 1.4 g/min of powder flow rate. Moreover, the samples production was carried out in argon atmosphere with O_2_ and H_2_O content of 1.2 ppm and 2.3 ppm, respectively. The alloyed powder was injected into the laser beam zone by high-purity argon (99.999%) which had a flow of 3 LPM. Due to avoiding the staircase effect, the contour for each layer was built as the first, and the next layers were deposited at angles of 30, 60, 120, and 150 degrees to the previous one. After fabrication, the cylindrical alloys samples were cut into slices with a thickness of 1 mm by the electrodischarge technique, which was included in epoxy resin and followed by mechanically grinding on SiC paper with granulations of 120, 240, 500, 1200, and 2400 in sequence and polished with 3 μm diamond suspensions. Finally, the obtained samples were removed from the resin. For each material variant, three testing samples were prepared. 

The anodization of the titanium alloys was performed in two types of electrolyte: glycol-based and glycerol-based solutions with 0.3 M NH4F and 2 wt % of deionized water addition. The deionized water content in the tested electrolyte types was selected based on the literature [14,17,18,20,31]. Before anodization, the sample surfaces were electropolished (electrochemical polishing on Struers LectroPol etching/electropolishing device, (Struers, Cleveland, OH, USA) in solution of perchloric acid, methanol, and butoxyethanol (A3 commercial electrolyte, Struers) at 30 V for 30 s. The anodization of all tested materials in both electrolytes was conducted at 50 V for 1 h and at 40 °C. A multimeter (RIGOL 3058E, Batronix, Preetz, Germany) was used to measure and store the registered current. The electrolytes viscosity was measured by IKA ROTAVISC lo-vi equipment (Staufen, Germany). The VMP-300 multichannel potentiostat form Bio-Logic Science Instruments (Seyssinet-Pariset, France) was used for the potentiodynamic polarization (PP) measurements of titanium alloys behavior in both types of tested electrolyte. A three-electrode cell assembly consisting of titanium alloy as the working electrode, stainless steel grid as the counter electrode, and a silver chloride electrode as the reference electrode was used. Before measuring the PP curves, the 1 h stabilization of the open circuit potential (OCP) was applied. The polarization curves were measured relative to the OCP. A scan rate of 1mV/s was applied, and a measurement point was taken every 0.2 s. 

Characterization of the substrates surface structure, anodic TiO_2_ morphology, and its chemical composition was performed using a scanning electron microscope Quanta 3D FEG (FEI, Hillsboro, OR, USA) equipped with an energy-dispersive X-ray spectroscopy (EDS) detector. The given chemical composition is averaged from three measurements collected from various places on samples each from areas of 600 × 600 µm^2^, at a working distance of 12 mm and at a magnification of 500×. The average nanopore diameter was estimated using ImageJ software (ver.1.53a, Madison, WI, USA). The oxide thickness was determined from SEM cross-sectional images.

## 3. Results and Discussion

Figure 1 shows the structure of the base and the referential Ti alloys substrate. LENS-made Ti alloys have the structure of martensite α’ (characteristic acicular martensite structure) (Figure 1a,c), which resulted from the quick cooling of the liquid metal pool (ca. 1 × 10^4^ °C/s) characteristic for the LENS process. The similar effect was observed by Chlebus et al. [32]. Moreover, the Ti6Al7Nb alloy has a significantly smaller grain size than the Ti6Al4V alloy. A referential commercially pure Ti alloy, in the form of thin foil, was delivered after cold rolling and annealing and characterized by the occurrence of equi-axial grains with diameters in the range of 20–30 µm (Figure 1b). One can see the crystal twinning in the referential samples structure (Figure 1b). However, it was recently found that the microstructure of the titanium surface not affect the anodic oxide nanotube diameter and length, which suggested that the other factors, such as chemical impurities or choice of anodizing electrolyte, have a significant influence on the morphological ATO parameters [31]. 

During the anodization of Ti alloys, the evolution of the current density as a function of time, which were named current curves, were registered (Figure 2). The average current density is presented in Table 1.

An average current density can be used to estimate the mass of substance deposited at the electrode during electrolysis according to Faraday’s law. It was found that the current density value depends on the type of electrolyte as well as the substrate type. Generally, a few times higher current density was observed during anodization of all the tested alloys conducted in glycol-based as compared to that in glycerine-based electrolyte (Figure 2c–e). For example, the average current density registered for the anodization of Ti6Al7Nb in glycol-based electrolyte was approximately seven times higher than for the similar process conducted in glycerine-based electrolyte (see Table 1). This phenomenon likely results from the difference in viscosity of the electrolytes solutions used. As published previously, in the case of aluminum anodization, the change in the viscosity of electrolyte is correlated with a change of the ionic mobility and influence on the morphological features of anodic aluminum oxide [20,22,23,24]. Moreover, the electrical conductivity of the electrolyte is inversely proportional to the viscosity, which is known as Walden’s empirical law [25]. The effect of electrolyte viscosity is even more spectacular in the case of titanium anodization [20]. In the present research, the viscosity of ethylene glycol-based electrolyte was 21.8 mPa·s, while for glycerine-based electrolyte, it was 94.1 mPa·s. Thus, the higher current density in glycol-based electrolyte is possible because of more than four times lower viscosity of this electrolyte compared to glycerine-based electrolyte. 

Furthermore, it was found that in one type of electrolyte, the current density depends on the substrate composition (Figure 2a,b, Table 1): the highest value was recorded for the Ti6Al7Nb alloy in both glycol-based and glycerine-based solutions. It is worth noticing that all the process parameters were the same in both used electrolytes, and only the type of alloy was different. The polarization measurements were conducted to investigate the differences in the electrochemical behavior of tested alloys in glycol- and glycerine-based electrolyte (Figure 3). For easy analysis, the polarization curves were broken down by electrolyte type (Figure 3a,b), and the tested materials (Figure 3c–e) are shown. Generally, it could be concluded that the lower the value of E, the easier the oxidation of materials. In Figure 3a, one can see that the polarization curves taken in glycol-based electrolyte are close to each other, and difference between the E values registered for Ti 99.5% and Ti6Al7Nb is slight. Only the polarization curve for Ti6Al4V is characterized by a significantly higher value of E. The E value differences of the polarization curves registered in glycerine-based electrolyte are more varied and increase in the following order: Ti6Al7Nb < Ti99.5% < Ti6Al4V (Figure 3b).

These results could be compared with a change of the average current density during anodization as well as the final oxide thickness, which changes in accordance with the same trend (Table 1). It is likely that the content of alloying elements in Ti alloys causes the observed differences in polarization curves measurements and behavior during anodizing. The investigations of the effect of substrate composition on anodic oxide morphology have been conducted for aluminum alloys for many years and mainly concern binary solid solution aluminum alloys [33,34,35,36,37]. In general, the anodic oxides can be doped in situ during anodization by both the ions of the electrolyte and also by the elements of the substrate material. The ions from the electrolyte are mainly incorporated into the anodic oxide and do not form nanoparticles. The exception may be e.g., hydrogen-bonded large molecules of dye which, due to their size, have limited ionic mobility in the electrolyte. They can agglomerate and/or the molecules could being firstly adsorbed on the growing oxide surface (on growing oxide/electrolyte interface), and then the expanding anodic oxide could enclose the adsorbed dopants [38,39,40]. In the case of Ti alloy anodization, it was found that the presence of alloying additives and chemical composition as well as phases composition of the Ti alloys substrate strongly affect the resulted anodic oxide morphology and composition [41,42]. Depending on the substrate chemical composition, the dissolution rate of the oxide layer is different and causes local variations of chemical composition as well as differences in grain structure and crystallography, in particular for multiphase alloys [14,43]. Moreover, the growth of anodic titanium oxide nanotubes at α and β Ti phases occurs at different speed and kinetics, which causes the formation of anodic layers with various heights [5]. In addition, the anodization of Ti alloys gives as a result an anodic oxide consisting of a mixture of oxides of alloying elements with different stoichiometry, depending on the used electrolyte [44]. Interestingly, it was found that the incorporation of species derived from the substrate occurring after the required enrichment of the alloy in the alloying element at the alloy/film interface is dependent upon the type of alloying element, and the degree of enrichment can be correlated with the Gibbs free energy per equivalent for formation of the alloying element oxide [34], which can be also compared to the galvanic series of elements. Moreover, for certain ternary and higher alloys, co-enrichments of alloying elements through anodic oxidation is possible, leading to anodic oxide films of relatively complex composition associated with different stages for oxidation of the individual alloying elements and differing migration rates of the oxidized species within the films [34]. According to the galvanic series, the susceptibility to oxidation increases as follows: V, Ti, Al, Nb, which corresponds with the results obtained in the presented research. Nb is an element that is more susceptible to oxidation than titanium and vanadium, so the susceptibility to oxidation of the Ti6Al7Nb alloy is higher than for the other ones. Based on the obtained results, one can see that the higher the content of elements that are more susceptible to oxidation in the substrate, the higher the average current density during anodizing and the higher the final oxide thickness. 

Polarization curves divided into tested materials are presented on the Figure 3c–e. The oxidation of Ti6Al4V alloy occurs similarly in both types of electrolytes (Figure 3c), while the oxidation of Ti 99.5% and Ti6Al7Nb alloys is preferred in glycol-based electrolyte (Figure 3d,e). Moreover, the difference in E values between both types of electrolyte is highest in the case of Ti6Al7Nb, which is reflected in the different resulting anodic oxide thicknesses: 10–11 µm and 3 µm in glycol-based and glycerine-based electrolyte, respectively (Table 1). In the case of Ti6Al4V, the thickness of anodic oxide made in both types of electrolyte is comparable (Table 1), which is in accordance with results of polarization. 

The morphology of fabricated anodic titanium oxide was nanotubular or nanoporous for all tested substrates and electrolytes (Figure 4). 

However, significant differences in the samples’ morphology were observed. In glycol-based electrolyte, the anodic titanium oxide (ATO) was produced in nanotubes form, while in glycerol-based electrolyte, a nanoporous ATO morphology was observed. Moreover, the nanotube walls obtained in the glycol-based electrolyte on the 99.5% Ti and Ti6Al7Nb alloy were partially dissolved at the top. It could be related to the current density and ionic mobility during anodization. When the current density is higher, the ionic mobility is higher, which results in higher anodization rates and as a consequence a higher dissolution of the ATO matrix. 

The anodization rate could be measured from the oxide growth rate and as such by comparison of the thickness of the oxide layers (see Table 1 and inserts in Figure 4). Dependent on the substrate type, a few times thicker ATO was formed in glycol-based than in glycerol-based electrolyte. The current density and thickness of the anodic oxide layer, which indicated an oxide growth rate, increased in the following order: Ti6Al4V < Ti 99.5% < Ti6Al7Nb for both types of electrolyte. The current density obtained during the anodization of Ti6Al4V, Ti 99.5%, and Ti6Al7Nb was 6.4 < 10.1 < 21.9 and 2.2 < 2.8 < 3.1 mA/cm^2^ respectively for glycol- and glycerine-based electrolyte. Simultaneously, the oxide nanotubes length fabricated on Ti6Al4V, Ti 99.5%, and Ti6Al7Nb was 1 < 7–8 < 10–11 and 0.7 < 2 < 3 µm respectively for glycol- and glycerine-based electrolyte. The thickness of anodic oxide produced on Ti6Al4V was relatively low for both tested electrolytes (Table 1). Moreover, the trend of pore diameter size is different. For both types of electrolyte, the pore diameter decreased as follows: Ti 99.5% > Ti6Al7Nb > Ti6Al4V, being 52, 48, 37 nm and 41, 27, 25 nm in glycol-based and glycerol-based electrolyte, respectively. It was expected that when the current density and anodic oxide growth rate are higher, the pore size will also be larger because the electrolyte penetrates deeper toward the anodized substrate. This is also the reason that nanotubes are formed: the walls of initial small and narrow nanopores are dissolved by the increased concentration of HF inside the pores, and nanotubes are generated [16,45]. When the dissolution of nanopore walls/nanotubes in the fluoride-containing electrolyte is intensive, the walls are almost completely dissolved, and a “nanograss”-type morphology at top area of the ATO layer could be observed [46,47,48]. In the present study, dissolution of the ATO top layer occurred for samples produced in the glycol-based electrolyte on Ti6Al7Nb and Ti 99.5% substrate. The different tendency of oxide pore diameter change compared to trends of current density changes and oxide growth rate could be caused by alloying elements, which are considered as impurities of pure titanium. As it was observed before in the case of aluminum substrate, in the same anodization conditions, the pore diameter of anodic aluminum oxide produced on high-purity Al was higher than on technical purity aluminum alloy [24]. The observation made in the presented research could be concluded as follows: the more impurities/alloying elements in substrate, the lower the pores diameters of its anodic oxide. However, the aim of this work is to find the difference among anodic oxides made on different titanium alloys prepared from glycol- and glycerol-based electrolytes rather than investigating the fundamental principles of the anodization mechanism. Additional research has to be done to clearly show the reason for the observed tendency of oxide pore diameter change.

For one substrate type, the chemical composition of the anodic oxide varied a lot with electrolyte type (Table 2). The content of main alloy elements in all tested alloys after anodization is comparable for both electrolyte types; however, the content of fluoride is strongly affected by electrolyte composition. The observed differences in chemical composition are much higher than the EDS method standard deviation for light elements (up to 1 at%). Much more fluoride was detected in samples made in glycerine-based electrolyte, even though the anodic oxide thickness was significantly smaller in that type of electrolyte (compare to Table 1). 

Generally, during anodization, ions from electrolyte are attracted to the electrodes and could be incorporated in growing anodic oxides [49,50]. In our study, we found that the fluoride anions preferably incorporate from glycerine-based electrolyte in the anodic matrix. As was published before, in the case of aluminum anodization, the differences in electrolyte viscosity influence the amount of incorporated electrolyte species into the anodic oxide [22,23,24]. In this research, similar to the case of aluminum anodization, the difference in the viscosity of the electrolyte strongly affected the chemical composition of anodic titanium oxide. As a result of the higher viscosity of the glycerine-based electrolyte, the fluoride anions are trapped near the anode longer than in the glycol-based electrolyte. Although the anodic oxide growth rate in glycerine-based electrolyte is much lower than that in glycol-based electrolyte, much more fluoride is incorporated into ATO fabricated from the electrolyte of increased viscosity. For example, the ATO produced on Ti6Al7Nb in glycol-based and glycerol-based electrolyte has a thickness of 10–11 µm and 3 µm, and the concentration of F is 11.00 at% and 15.12 at%, respectively (compare Table 1 and Table 2). Since the chemical composition of materials strongly influences its physicochemical properties, our findings could be used in the future to control, for example, the surface functionalization or photocatalytical properties of anodic titanium oxides. 

## 4. Conclusions

After anodization in glycol-based as well as glycerol-based electrolyte, a nanoporous/nanotubular morphology of anodic titanium oxide on Ti6Al7Nb, Ti 99.5%, and Ti6Al4V alloys surface was obtained. The current density and oxide growth rate are affected by the used electrolyte as well as by substrate chemical compositions. Generally, the oxide growth rate is higher in glycol-based than in glycerol-based electrolyte for all tested materials. It was found that the substrate composition has a significant impact on anodic oxide. For one type of electrolyte, the highest current density and oxide growth rate is for Ti6Al7Nb alloy, and the smallest is for Ti6Al4V alloy. It is probably due to the alloy composition, because niobium is an element that is more susceptible to oxidation than titanium and vanadium, so susceptibility to the oxidation of Ti6Al7Nb is higher than that for the other tested alloys. It was found that the amount of incorporated ions into the anodic matrix strongly depended on the electrolyte viscosity, which can be used in the future design of smart materials. The amount of fluoride in ATO made in glycerol-based electrolyte was a few times higher compared to that in ATO produced in glycol-based electrolyte for the same substrate. 

## Figures and Tables

**Figure 1 materials-13-04743-f001:**
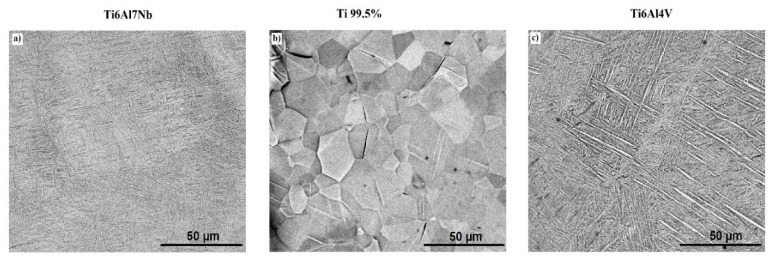
The structure of the base (Ti6Al7Nb (**a**) and Ti6Al4V (**c**)) and the referential Ti (Ti 99.5% (**b**)) alloys substrate.

**Figure 2 materials-13-04743-f002:**
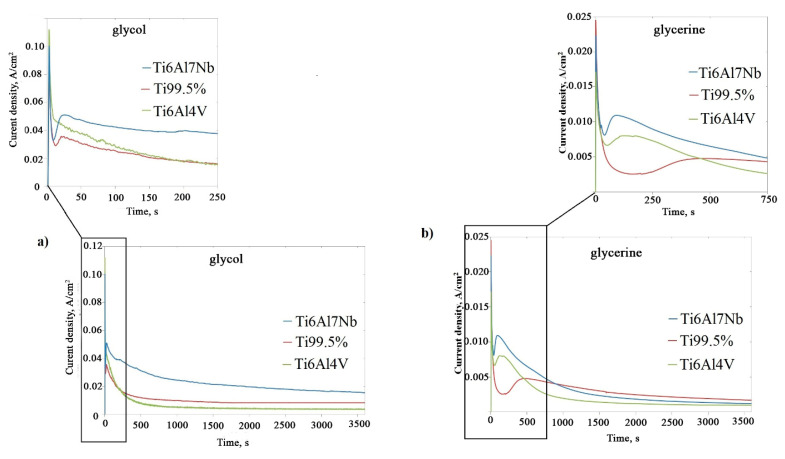
Current density as a function of anodization time (the current curves) for anodization in glycol-based (**a**) and glycerol-based (**b**) electrolyte. The current curves for individual alloy compositions: Ti6Al7Nb (**c**), Ti 99.5% (**d**), and Ti6Al4V (**e**).

**Figure 3 materials-13-04743-f003:**
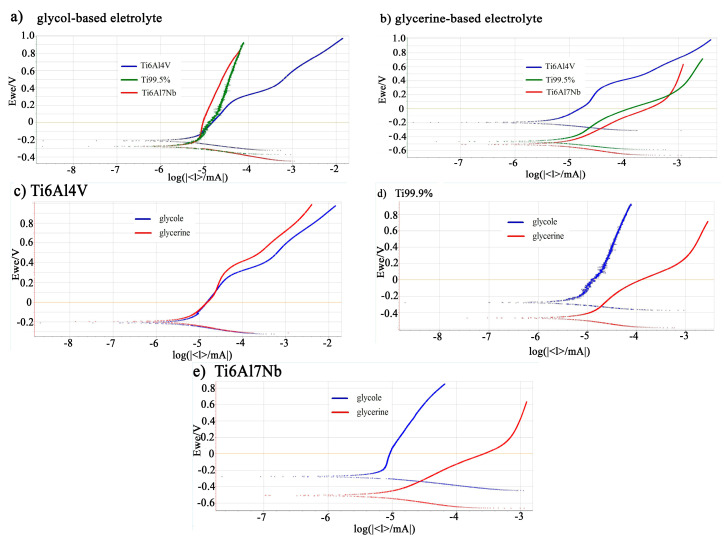
Polarization curves measured in glycol-based (**a**) and glycerine-based (**b**) electrolyte. The polarization curves for individual alloy composition: Ti6Al4V (**c**), Ti 99.5% (**d**), and Ti6Al7Nb (**e**).

**Figure 4 materials-13-04743-f004:**
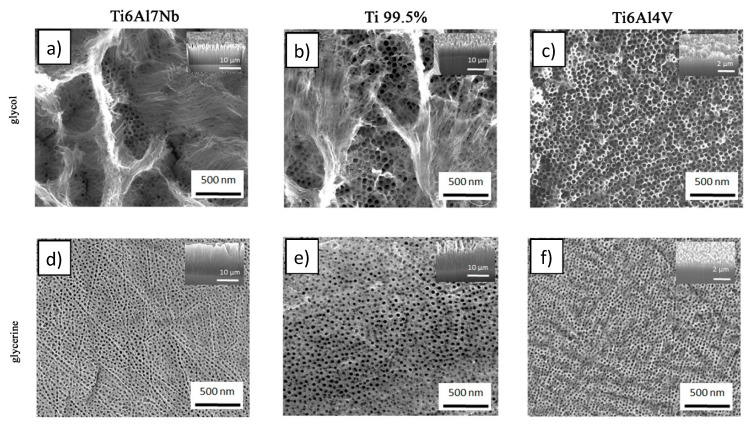
Top view and cross-section (insert) of anodic oxide layers on Ti6Al7Nb, Ti 99.5%, and Ti6Al4V alloys by anodization in glycol-based (**a**–**c**) and glycerol-based (**d**–**f**) electrolyte. In the top view, the scale bar corresponds to 500 nm for every substrate. In the cross-section, the scale bar represents 10 µm for Ti6Al7Nb and Ti 99.5%, and 2 µm for Ti6Al4V.

**Table 1 materials-13-04743-t001:** Maximum and average current density, and titanium anodic oxide morphological features depending of the substrate and electrolyte composition.

	Ti6Al7Nb		Ti 99.5%		Ti6Al4V	
	Glycol	Glycerine	Glycol	Glycerine	Glycol	Glycerine
Maximum Current Density [mA/cm^2^]	98.3	22.1	78.1	24.3	108.9	17.0
Average Current Density [mA/cm^2^]	21.9	3.1	10.1	2.8	6.4	2.2
Oxide Thickness [µm]	10–11	3	7–8	2	1	>1
Nanotube/Nanopore Diameter [nm]	48	27	52	41	37	25

**Table 2 materials-13-04743-t002:** Chemical composition (at%) of tested alloys before and after anodization in glycol- and glycerine-based electrolyte measured by the EDS method.

	Ti6Al7Nb			Ti 99.5%			Ti6Al4V		
	Before Anodization	After Anodization		Before Anodization	After Anodization		Before Anodization	After Anodization	
		Glycol	Glycerine		Glycol	Glycerine		Glycol	Glycerine
**Ti**	84.6	32.1	33.9	89.0	33.7	41.2	84.9	44.4	41.7
**Al**	11.7	3. 5	3.8	–	–	–	12.5	5.7	5.4
**Nb**	3.8	2.2	1.9	–	–	–	–	–	–
**V**	–	–	–	–	–	–	2. 6	1.4	1.5
**O**	–	50.3	44. 3	–	52.4	42.7	–	35.0	35.6
**F**	–	11.0	15.1	–	9.2	10.5	–	5.8	10.3
**C**	–	0.9	1.0	–	4.7	5.6	–	7.7	5.3

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
