# Peer review of "Self-Organized Anodic Oxides on Titanium Alloys Prepared from Glycol- and Glycerol-Based Electrolytes"

_materials, 2020, doi:10.3390/ma13214743_

Round 1

Reviewer 1 Report

The article is devoted to the study of anodic oxidation of titanium alloys. The subject of the work is quite interesting because the resulting materials have a wide range of applications. In General, the article makes a positive impression. The authors conducted a detailed literature review, and the experiment is described quite carefully. I believe that this work can be published in this journal.
However, I would like to ask the authors to improve figures 2 and 3. It is necessary to use a better resolution of the illustrations and increase the data captions.

Reviewer 2 Report

The authors deal with the formation of self-organized anodic oxides on two types of titanium alloys.

Although the anodization of these two alloy was fully investigated in a previous study (), using the two different electrolytes can be the novelty of their work. However, before publishing the manuscript there some major points that should be considered:

  • From the results, it seems that no optimization of the electrolyte has been done to grow self organized nanotubes. I believe that by varying the water content in the solution one can modify the nanoporous to nanotubes.
  • There is no discussion or result about the nature of the secondary and ternary elements in the alloy after the anodization.

        Are the elements form nanoparticle in the structure or are they doped in the oxide?

Reviewer 3 Report

Dear authors,

You did an interesting work but the presentation of it does not satisfy. My remarks are as follows:

In Abstract, abbreviation (LENS) should not be given.

P3 L106-109 In experimental part, the alloys production process should be described in more details.

               It is not perfectly clear the exact number of samples. Was there only one sample of each alloy?

What was the metal powders purity and the morphology?

Which were the conditions of LENS (such as laser power, scan speed, powder feed rate, temperature, time etc.) and how they were selected/determined? How the cooling was carried out? What there was taken to avoid a staircase effect?

P3 L111-113 How are the samples metallographically prepared? Whether the samples are embedded in the epoxy resin for grinding and to what gradation of SiC paper were they grinded?

In the caption of table 2 should be stated that the results were obtained by EDS method. Since EDS method is semi-quantitative, 2 decimals have no sense. What was the standard deviations? In which % the composition is expressed in this table?

Round 2

Reviewer 2 Report

The author's reply to the second comment is not correct, it means that doping or formation of the nanoparticles on tubes depends on the secondary elements (noble or not noble). The present study is on the anodization of Ti alloys, therefore the other alloying elements cannot be ignored in the investigation. Therefore, I recommend the publication of the manuscript after elucidating the point in the manuscript text and adding further explanation regarding the nature of the secondary and ternary materials. 

Author Response

Dear Reviewer,

Please see the attachment file.

Kind regards,

Authors
